# CAFE: Catastrophic Data Leakage in Federated Learning

## Abstract

Private training data can be leaked through the gradient sharing mechanism deployed in machine learning systems, such as federated learning (FL). Increasing batch size is often viewed as a promising defense strategy against data leakage. In this paper, we revisit this defense premise and propose an advanced data leakage attack to efficiently recover batch data from the shared aggregated gradients. We name our proposed method as *catastrophic data leakage in federated learning (CAFE)*. Comparing to existing data leakage attacks, CAFE demonstrates the ability to perform large-batch data leakage attack with high data recovery quality. Experimental results on vertical and horizontal FL settings have validated the effectiveness of CAFE in recovering private data from the shared aggregated gradients. Our results suggest that data participated in FL, especially the vertical case, have a high risk of being leaked from the training gradients. Our analysis implies unprecedented and practical data leakage risks in those learning settings.

## 1 Introduction

Federated learning (FL) (Chilimbi et al., 2014; Shokri & Shmatikov, 2015) is an emerging machine learning framework where a central server and multiple workers collaboratively train a machine learning model. Most of existing FL methods consider the setting where each worker has data of a different set of subjects but their data share many common features. This setting is also referred to data partitioned or horizontal FL (HFL). Unlike the HFL setting, in many learning scenarios, multiple workers handle data about the same set of subjects, but each has a different set of features. This case arises in financial and healthcare applications (Chen et al., 2020). In these examples, data owners (e.g., financial institutions and hospitals) have different records of those users in their joint user base, so, by combining their features, they can establish a more accurate model. We refer to this setting as feature-partitioned or vertical FL (VFL).

Compared with existing distributed learning paradigms, FL raises new challenges including the heterogeneity of data and the privacy of data (McMahan et al., 2017). To protect data privacy, only model parameters and the change of parameters (e.g., gradients) are exchanged between server and workers (Li, 2014; Iandola et al., 2015). Recent works have studied how a malicious worker can embed backdoors or replace the global model in FL (Bagdasaryan et al., 2018; Bhagoji et al., 2019; Xie et al., 2020). As exchanging gradients is often viewed as privacy-preserving protocols, little attention has been paid to information leakage from public shared gradients and batch identities.

In this context, inferring private user data from the gradients has received growing interests (Fredrikson et al., 2015; Hitaj et al., 2017; Melis et al., 2018). A popular method that was termed deep leakage from gradients (DLG) has been developed in (Zhu et al., 2019) that infers training data in an efficient way without using any generative models or prior information. However, DLG lacks generalizability on model architecture and weight distribution initialization (Geiping et al., 2020). In Zhao et al. (2020), an analytical approach has been developed to extract accurate labels from the gradients. Wang et al. (2020) proposed a novel gradient difference as a distance measure to improve recovery accuracy. However, all of them cannot scale up to the large-batch data leakage setting.

The contributions of this paper are summarized in the following.

1) We develop an advanced data leakage attack that we term CAFE to overcome the limitation of current data leakage attacks on FL. CAFE is able to recover large-scale data both in VFL and HFL.

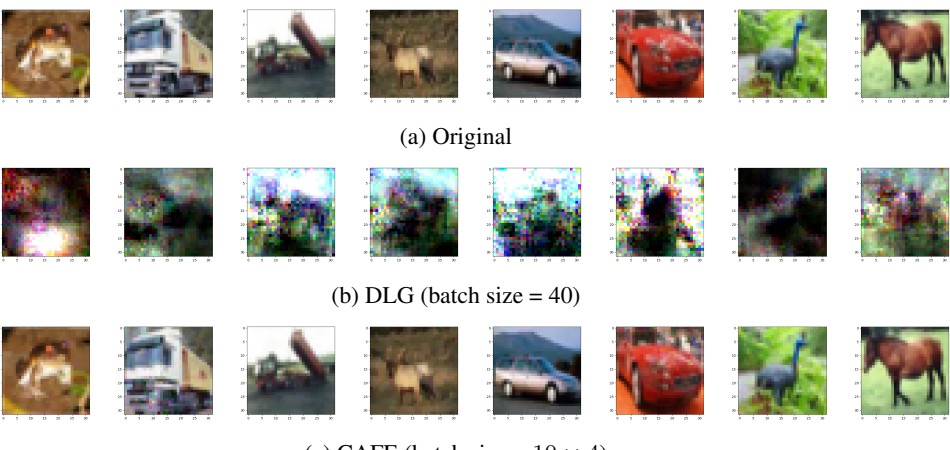

(a) Original

(b) DLG (batch size = 40)

(c) CAFE (batch size = $10 \times 4$)

Figure 1: Illustration of large-batch data leakage on CIFAR-10 from shared gradients in FL

2) Our large-batch data recovery is based on the novel use of data index alignment and internal representation alignment in FL, which can significantly improve the recovery performance.

3) The effectiveness and practical risk induced from our data leakage algorithm is justified in the dynamic FL training setting when all parameters in the model are updated every iteration.

## 2 PRELIMINARY

FL can be categorized into horizontal and vertical FL settings (Kairouz et al., 2019). In this section, we provide necessary background of FL in this section.

**Horizontal FL.** In HFL, data are distributed among local workers holding the same feature space. Suppose that there are $M$ workers participating in the FL process and the number of samples in the dataset $\mathcal{X}$ is $N$. The dataset is denoted as $\mathcal{X} := [\mathbf{X}_1, \ldots, \mathbf{X}_m, \ldots, \mathbf{X}_M]^T$, where $\mathbf{X}_m \in \mathbb{R}^{N_m \times p}$ is the local data partitioned to worker $m$, and $p$ is the dimension of data feature space, $N_m$ is the number of data samples partitioned to local worker $m$, and $\sum_{m=1}^{M} N_m = N$. Since all local data share the same feature space, each local worker computes the gradients independently and uploads them to the server. The server receives all gradients from each local worker and uses gradient aggregation methods such as FedAvg (Konečný et al., 2016). Let the parameters of the model as $\boldsymbol{\theta}$ and the loss function as $\mathcal{L}$. Then the objective function of HFL can be defined as:

$$\min_{\boldsymbol{\theta}} \quad \frac{1}{N} \sum_{m=1}^{M} \mathcal{L}(\boldsymbol{\theta}; \mathbf{X}_m) \quad \text{with} \quad \mathcal{L}(\boldsymbol{\theta}; \mathbf{X}_m) := \sum_{n \in \mathcal{N}_m} \mathcal{L}(\boldsymbol{\theta}; \mathbf{x}_n) \tag{1}$$

**Vertical FL.** Different from HFL, in VFL, each local worker $m$ is associated with a unique set of features. Each data sample $\mathbf{x}_n$ in dataset $\mathcal{X}$ can be written as

$$\mathbf{x}_n = [\mathbf{x}_{n1}^T, \ldots, \mathbf{x}_{nm}^T, \ldots, \mathbf{x}_{nM}^T]^T \tag{2}$$

where $\mathbf{x}_{nm} \in \mathbb{R}^{p_m}$ is the data partitioned to worker $m$ and $p_m$ is the data dimension in local worker $m$. The label space $\{y_n\}_{n=1}^{N}$ can be regarded as a special feature and is partitioned to the server or a certain local worker. Similar to (1), the objective function of VFL can be written as:

$$\min_{\boldsymbol{\theta}} \quad \frac{1}{N} \sum_{n=1}^{N} \mathcal{L}(\boldsymbol{\theta}; \mathbf{x}_{n1}; \ldots; \mathbf{x}_{nM}) \tag{3}$$

## 3 CATASTROPHIC DATA LEAKAGE FROM BATCH GRADIENTS

To realize large-scale data recovery from aggregated gradients, we propose our algorithm named as *CAFE: Catastrophic dAta leakage in Federated lEarning*. While CAFE can be applied to any type of data, without loss of generality, we use image datasets throughout the paper.

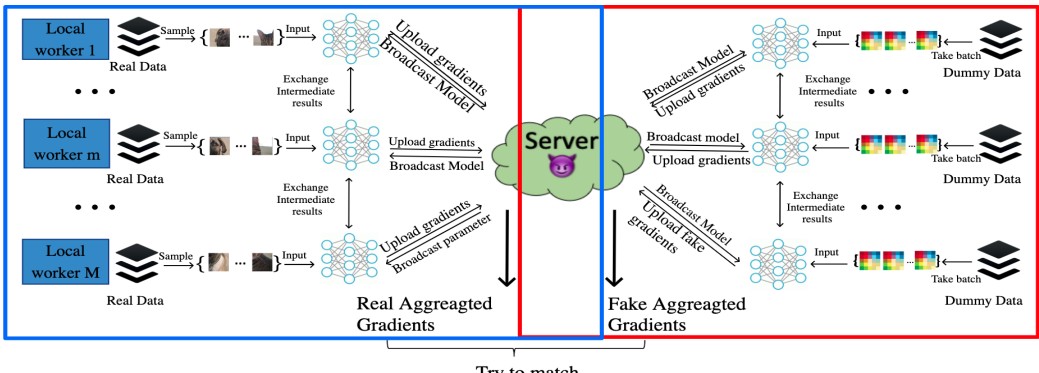

Figure 2: Overview of CAFE in VFL

### 3.1 WHY LARGE-BATCH DATA LEAKAGE ATTACK IS DIFFICULT?

We start by providing some intuition on the difficulty of performing large-batch data leakage from aggregated gradients based on the formulation of DLG (Zhu et al., 2019). Assume that $N$ images are selected as the input for a certain learning iteration. We define the data batch as $\mathcal{X} = \{\mathbf{x}_n, y_n | \mathbf{x}_n \in \mathbb{R}^{H \times W \times C}, n = 1, 2, \ldots, N\}$, where $H, W, C$ represents the height, the width and the channel number of each image. Likewise, the batched 'recovered data' is denoted by $\hat{\mathcal{X}} = \{\hat{\mathbf{x}}_n, \hat{y}_n | \hat{\mathbf{x}}_n \in \mathbb{R}^{H \times W \times C}, n = 1, 2, \ldots, N\}$, which have the same dimension as $\mathcal{X}$. Then the objective function is

$$\hat{\mathcal{X}}^* = \arg\min_{\hat{\mathcal{X}}} \left\| \frac{1}{N} \sum_{n=1}^{N} \nabla_{\boldsymbol{\theta}} \mathcal{L}(\boldsymbol{\theta}, \mathbf{x}_n, y_n) - \frac{1}{N} \sum_{n=1}^{N} \nabla_{\boldsymbol{\theta}} \mathcal{L}(\boldsymbol{\theta}, \hat{\mathbf{x}}_n, \hat{y}_n) \right\|^2 \tag{4}$$

Note that in (4), the dimensions of the aggregated gradients is fixed. However, as the $N$ increases, the dimension of $\hat{\mathcal{X}}$ and $\mathcal{X}$ rise. When $N$ is sufficiently large, it will be more challenging to find the "right" solution $\hat{\mathcal{X}}$ of (4) corresponding to the ground-truth dataset $\mathcal{X}$. On the other hand, CAFE addresses this large-batch issue by data index alignment for batch data recovery, which can effectively exclude undesired solutions. We discuss a specific example in Appendix A.

As a motivating example, Figure 1 compares our proposed attack with DLG on a batch of 40 images. The recovery quality of DLG is far from satisfactory, while CAFE can successfully recover all images in the batch. It is worth noting that because DLG is not effective on large-batch recovery, it is suggested in Zhu et al. (2019) that increasing batch size could be a promising defense. However, the successful recovery of CAFE shows that such defense premise gives a false sense of security in data leakage and the current FL is at risk, as large-batch data recovery can be accomplished.

### 3.2 CAFE IN VFL

In VFL, the server sends public key to local workers and decides the data index in each iteration of training and evaluation (Yang et al., 2019; Cheng et al., 2019). During the training process, local workers exchange their intermediate results with others to compute gradients and upload them. Therefore, the server has the access to both the model parameters and their gradients. Notably, CAFE can be readily applied to existing VFL protocols where the batch data index is assigned.

Figure 2 gives an overview of CAFE in the VFL setting. The blue part represents a normal VFL paradigm and the red part represents the CAFE attack. Since data are vertically partitioned among different workers, data index alignment turns out to be an inevitable step in the vertical training process, which provides the server (the attacker) an opportunity to control the selected batch data index. Suppose that there are $M$ workers participating FL and the batch size is $N$. The aggregated gradients can be denoted by

$$\nabla_{\boldsymbol{\theta}} \mathcal{L}(\boldsymbol{\theta}, \mathcal{X}^t) = \frac{1}{N_b} \sum_{n=1}^{N_b} \nabla_{\boldsymbol{\theta}} \mathcal{L}(\boldsymbol{\theta}, \mathcal{X}_n^t) \quad \text{with} \quad \mathcal{X}_n^t = [\mathbf{x}_{n1}^t, \mathbf{x}_{n2}^t, \ldots, \mathbf{x}_{nM}^t]. \tag{5}$$

---

**Algorithm 1** CAFE in VFL ( regular VFL protocol and CAFE protocol )

---

1: Initialize model parameters $\boldsymbol{\theta}$ and generate fake data $\hat{\mathcal{X}}$
2: **for** $t = 1, 2, \ldots, T$ **do**
3:     Server broadcasts the global model to all local workers (a total of $M$ workers)
4:     **for** $m = 1, 2, \ldots, M$ **do**
5:         Worker $m$ takes real batch data
6:         Worker $m$ computes the intermediate results and exchanges them with other workers
7:         Worker $m$ uses the exchanged intermediate results to compute local aggregated gradients
8:         Worker $m$ uploads real local aggregated gradients to the server.
9:     **end for**
10:    Server computes real global aggregated gradients $\nabla_{\boldsymbol{\theta}} \mathcal{L}(\boldsymbol{\theta}, \mathcal{X}^t)$
11:    Server computes the fake global aggregated gradients $\nabla_{\boldsymbol{\theta}} \mathcal{L}(\boldsymbol{\theta}, \hat{\mathcal{X}}^t)$
12:    Server computes CAFE loss: $\mathbb{D}(\mathcal{X}^t; \hat{\mathcal{X}}^t)$ and $\nabla_{\hat{\mathcal{X}}^t} \mathbb{D}(\mathcal{X}^t; \hat{\mathcal{X}}^t)$
13:    Server updates the batch data $\hat{\mathcal{X}}^t$ with $\nabla_{\hat{\mathcal{X}}^t} \mathbb{D}(\mathcal{X}^t; \hat{\mathcal{X}}^t)$
14:    Server updates the model parameters $\boldsymbol{\theta}$ with $\nabla_{\boldsymbol{\theta}} \mathcal{L}(\boldsymbol{\theta}, \mathcal{X}^t)$
15: **end for**

---

A benign server will perform legitimate computations designed by FL protocol. However, as shown in Figure 2, a curious server can provide the same legitimate computation as a benign server while simultaneously perform data recovery in a stealthy manner. The server symmetrically generates fake images corresponding to the real ones. Once a batch of original data is selected, the server takes the corresponding fake batch and obtains the fake gradients as

$$\nabla_{\boldsymbol{\theta}} \mathcal{L}(\boldsymbol{\theta}, \hat{\mathcal{X}}^t) = \frac{1}{N_b} \sum_{n=1}^{N_b} \nabla_{\boldsymbol{\theta}} \mathcal{L}(\boldsymbol{\theta}, \hat{\mathcal{X}}_n^t) \quad \text{with} \quad \hat{\mathcal{X}}_n^t = [\hat{\mathbf{x}}_{n1}^t, \hat{\mathbf{x}}_{n2}^t, \ldots, \hat{\mathbf{x}}_{nM}^t]. \tag{6}$$

Algorithm 1 gives a pseudo code that implements our CAFE attack in VFL cases. The key part in our algorithm is aligning the real data batch indices with the fake ones. We define the squared $\ell_2$-norm of the difference between the real and fake aggregated gradients in (7). Since the server has the access to the model parameters, the attacker is able to compute the gradient of fake data from the loss in (7) and optimize the fake data for the purpose of recovering real data.

$$\mathbb{D}(\mathcal{X}^t; \hat{\mathcal{X}}^t) = \left\| \nabla_{\boldsymbol{\theta}} \mathcal{L}(\boldsymbol{\theta}, \mathcal{X}^t) - \nabla_{\boldsymbol{\theta}} \mathcal{L}(\boldsymbol{\theta}, \hat{\mathcal{X}}^t) \right\|^2. \tag{7}$$

### 3.3 AUXILIARY REGULARIZERS

In addition to the gradient matching loss in (7), we further introduce two regularization terms – *internal representation regularization* and *total variance (TV) norm*. Motivated by (Geiping et al., 2020), the input vectors of the first fully connected layer can be directly derived from the gradients, we define the real/fake inputs of the first fully connected layer at the $t$th iteration as $\mathcal{Z}^t / \hat{\mathcal{Z}}^t \in \mathbb{R}^{N \times P}$ and we use $\ell_2$ norm of their difference as what we call internal representation regularization.

To promote the smoothness of the fake images, we assume the TV norm of the real images as a constant, $\xi$, and compare it with the TV norm of the fake ones, $TV(\hat{\mathcal{X}})$. For each image $\mathbf{x} \in \mathbb{R}^{H \times W \times C}$ in data batch $\mathcal{X}^t$, its TV norm is denoted by $TV(\mathbf{x}) = \sum_c \sum_{h,w} \left[ |\mathbf{x}_{h+1,w,c} - \mathbf{x}_{h,w,c}| + |\mathbf{x}_{h,w+1,c} - \mathbf{x}_{h,w,c}| \right]$. As the result, the loss at the $t$th iteration $\mathbb{D}(\mathcal{X}^t, \hat{\mathcal{X}}^t)$ can be rewritten as:

$$\mathbb{D}(\mathcal{X}^t; \hat{\mathcal{X}}^t) = \left\| \nabla_{\boldsymbol{\theta}} \mathcal{L}(\boldsymbol{\theta}, \mathcal{X}^t) - \nabla_{\boldsymbol{\theta}} \mathcal{L}(\boldsymbol{\theta}, \hat{\mathcal{X}}^t) \right\|^2 + \beta TV(\hat{\mathcal{X}}) \cdot \mathbb{1}_{\{TV(\hat{\mathcal{X}}) - \xi \geq 0\}} + \gamma \left\| \mathcal{Z}^t - \hat{\mathcal{Z}}^t \right\|_F^2, \tag{8}$$

where $\beta$ and $\gamma$ are coefficients and $\mathbb{1}_{\{TV(\hat{\mathcal{X}}) - \xi \geq 0\}}$ is the indicator function. We will provide an ablation study in Section 4.5 to demonstrate the utility of these regularizers.

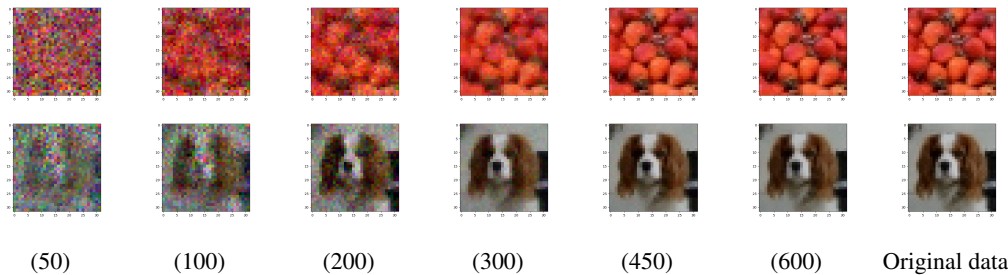

(50)   (100)   (200)   (300)   (450)   (600)   Original data

Figure 3: CAFE on Linnaeus
(Epoch: 50, 100, 200, 300, 450, 600, Original data)

### 3.4 CAFE IN HFL

Similarly, we can apply our CAFE algorithm to HFL as well. Let $\mathcal{X}_m^t$ denote the original batch data taken by local worker $m$ at the $t$th iteration. The gradients of the parameters at the $t$th iteration is

$$\nabla_{\boldsymbol{\theta}}\mathcal{L}(\boldsymbol{\theta}, \mathcal{X}^t) = \frac{1}{M}\sum_{m=1}^{M}\nabla_{\boldsymbol{\theta}}\mathcal{L}(\boldsymbol{\theta}, \mathcal{X}_m^t)\,, \mathcal{X}^t = \{\mathcal{X}_1^t, \mathcal{X}_2^t, \dots, \mathcal{X}_m^t, \dots, \mathcal{X}_M^t\}. \tag{9}$$

Symmetrically, we define the batch fake data and fake aggregated gradients as

$$\nabla_{\boldsymbol{\theta}}\mathcal{L}(\boldsymbol{\theta}, \hat{\mathcal{X}}^t) = \frac{1}{M}\sum_{m=1}^{M}\nabla_{\boldsymbol{\theta}}\mathcal{L}(\boldsymbol{\theta}, \hat{\mathcal{X}}_m^t)\,, \hat{\mathcal{X}}^t = \{\hat{\mathcal{X}}_1^t, \hat{\mathcal{X}}_2^t, \dots, \hat{\mathcal{X}}_m^t, \dots, \hat{\mathcal{X}}_M^t\}. \tag{10}$$

Due to space limitation, we will provide the CAFE algorithm for HFL in Appendix B.

## 4 PERFORMANCE EVALUATION

### 4.1 EXPERIMENT SETUPS AND DATASETS

We conduct experiments on CIFAR-10, CIFAR-100 and Linnaeus 5 datasets in both HFL and VFL settings. All the fake data are initialized uniformly and optimized by the normalized gradient descent method. Our algorithm can recover all the data participating in FL with a relative large batch size (more than $40$). Scaling up to our hardware limits, CAFE can leak as many as 2000 images in the VFL setting including $4$ workers.

**Evaluation metrics.** To measure the data leakage performance, we introduce peak signal-to-noise ratio (PSNR) value with mean squared error (MSE) defined in (11) and (12). Higher PSNR value of leaked data represents better performance of data recovery.

$$\text{MSE}_c(\mathbf{x}, \hat{\mathbf{x}}) = \frac{1}{HW}\sum_{i=1}^{H}\sum_{j=1}^{W}[\mathbf{x}_{ijc} - \hat{\mathbf{x}}_{ijc}]^2 \tag{11}$$

$$\text{PSNR}(\mathbf{x}, \hat{\mathbf{x}}) = \frac{1}{C}\sum_{c=1}^{C}\left[20\log_{10}(\max_{i,j}\mathbf{x}_{ijc}) - 10\log_{10}(\text{MSE}_c(\mathbf{x}, \hat{\mathbf{x}}))\right]. \tag{12}$$

**Baseline methods for comparison.** We compare CAFE with three other baselines, (i) DLG (Zhu et al., 2019), (ii) DLG given labels (iDLG) (Zhao et al., 2020), and (iii) using cosine similarity to compare the real and fake gradients (Geiping et al., 2020). We implement the original DLG and our CAFE under the same model and optimization methods. We run the DLG on 50 single images respectively and compute the average iterations required to make the PSNR value of a single leaked image above 30. We also compute the expected iteration number per image leakage for our CAFE algorithm. Furthermore, we fix the batch size, and compare the PSNR value obtained by CAFE with that of DLG. We also test the impact of given labels on CAFE by using the techniques in (Zhao et al., 2020). Moreover, we compare the performance of CAFE under different loss functions: i) replacing the squared $\ell_2$ norm term with the cosine similarity of two gradients (CAFE with cosine similarity) ii) loss proposed in (Geiping et al., 2020), which only contains the TV norm regularizer.

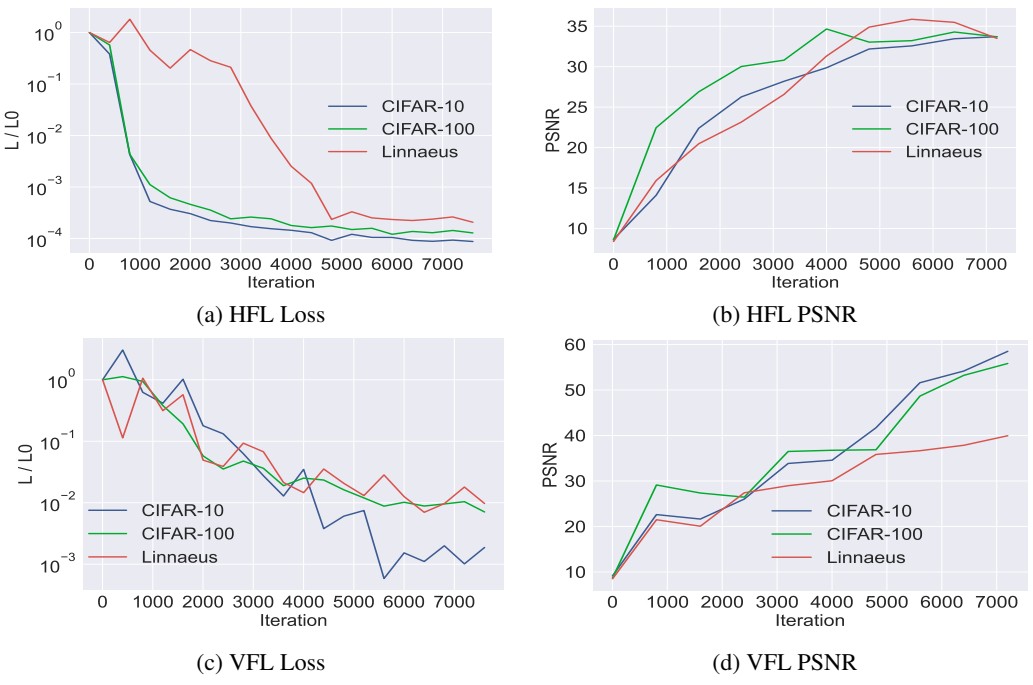

Figure 4: CAFE loss ratio and PSNR curves

Table 1: CAFE vs DLG

| Iterations ＼ Datasets Batch size | CIFAR-10 | CIFAR-100 | Linnaeus |
|---|---|---|---|
| 1(DLG) | 284.4 | 266.9 | 366.7 |
| $10 \times 4$ | 9.50 | 6.00 | 9.50 |
| $20 \times 4$ | 6.75 | 3.86 | 4.75 |
| $30 \times 4$ | 4.83 | 3.41 | 3.17 |
| $40 \times 4$ | 3.75 | 3.75 | 2.375 |

(a) Comparison of data leakage speed. Lower iteration count is faster.

| PSNR ＼ Datasets Algorithm | CIFAR-10 | CIFAR-100 | Linnaeus |
|---|---|---|---|
| CAFE | 35.03 | 36.90 | 36.37 |
| DLG | 10.09 | 10.79 | 10.10 |

(b) Comparison of leakage performance. Higher PSNR is better. Batch size = 40.

Table 2: PSNR via Loss

| PSNR ＼ Datasets Loss | CIFAR-10 | CIFAR-100 | Linnaeus |
|---|---|---|---|
| CAFE ((8)) | 35.03 | 36.90 | 36.37 |
| CAFE with cosine similarity | 30.15 | 31.38 | 30.76 |
| Loss in (Geiping et al., 2020) | 16.95 | 19.74 | 16.42 |

(a) HFL
(4 workers, batch ratio $= 0.1$, batch size $10 \times 4$)

| PSNR ＼ Datasets Loss | CIFAR-10 | CIFAR-100 | Linnaeus |
|---|---|---|---|
| CAFE ((8)) | 43.31 | 48.10 | 35.06 |
| CAFE with cosine similarity | 30.96 | 43.68 | 34.90 |
| Loss in (Geiping et al., 2020) | 12.76 | 10.85 | 10.46 |

(b) VFL
(4 workers, batch ratio $= 0.1$, batch size $40$)

## 4.2 CAFE IN HFL SETTINGS

In the HFL setting, we use a neural network consisting of 2 convolutional layers and 3 fully connected layers. The number of output channels of the convolutional layers are 64 and 128 respectively. The number of nodes of the first two fully connected layers are 512 and 256. The last layer is the softmax classification layer. We assume that 4 workers are involved in HFL and each of them holds a dataset including 100 images. The batch size of each worker in the training is 10, so there are 40 $(10 \times 4)$ images in total participating per iteration. For each experiment, we initialize the fake data using uniform distribution and optimize them for 800 epochs.

Figures 4a and 4b show the CAFE loss curves and the PSNR curves on the three datasets in HFL cases. In the loss ratio curve, we set the ratio of current CAFE loss and the initial CAFE loss $\frac{\mathcal{L}(\boldsymbol{\theta}, \mathcal{X}^t)}{\mathcal{L}(\boldsymbol{\theta}, \mathcal{X}^0)}$ as label $y$. The PSNR values are always above 35 at the end of each CAFE attacking process, suggesting high data recovery quality (see Figure 1 as an example). Figure 3 shows the attacking process of CAFE on Linnaeus. Under CAFE, PSNR reaches 35 at the 450th epoch where the private training data are completely leaked visually.

**Comparison with DLG baseline.** In Table 1a, we set the batch ratio in CAFE as 0.1 and compare it with DLG under different batch sizes. Clearly, CAFE outperforms DLG thanks to our novel design of large-batch data leakage attack. As shown in Table 1b, DLG cannot obtain satisfactory results when the batch size increases to 40, while CAFE successfully recovers all the images. Due to similarity between iDLG and DLG, the results are in Appendix C.

**Comparison with cosine similarity.** Table 2a shows that the PSNR values are still above 30 if we use cosine similarity instead of $\ell_2$ norm. The slight drop in PSNR value may result from scaling ambiguity in cosine similarity. There is a performance gap between the loss of CAFE and the loss in Geiping et al. (2020), which validates the importance of our proposed auxiliary regularizers.

Table 3: PSNR via Batch size

| PSNR \ Datasets Batch size | CIFAR-10 | CIFAR-100 | Linnaeus |
|---|---|---|---|
| 10 per worker | 35.03 | 36.90 | 36.37 |
| 20 per worker | 33.14 | 33.99 | 36.32 |
| 30 per worker | 32.31 | 33.21 | 35.96 |
| 40 per worker | 30.59 | 30.70 | 35.49 |

(a) HFL (4 workers, batch ratio = 0.1)

| PSNR \ Datasets Batch size | CIFAR-10 | CIFAR-100 | Linnaeus |
|---|---|---|---|
| 8 | 41.80 | 44.42 | 39.96 |
| 40 | 59.51 | 65.00 | 41.37 |
| 80 | 57.20 | 63.10 | 43.66 |
| 160 | 54.74 | 64.75 | 38.72 |

(b) VFL (4 workers, batch ratio = 0.2)

Table 4: PSNR via Batch ratio

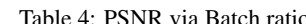

| PSNR \ Datasets Batch ratio | CIFAR-10 (HFL) | Linnaeus | CIFAR-10 (VFL) |
|---|---|---|---|
| 0.1 | 34.10 | 35.38 | 48.78 |
| 0.05 | 34.49 | 32.92 | 55.46 |
| 0.02 | 37.96 | 35.66 | 48.45 |
| 0.01 | 35.39 | 36.56 | 46.46 |

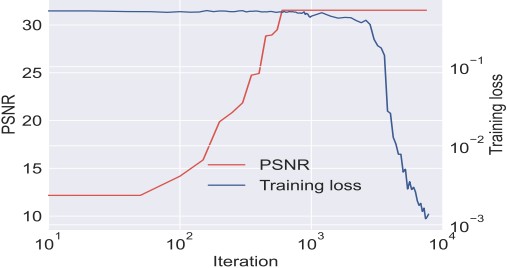

Figure 5: PSNR and training loss curves

### 4.3 CAFE IN VFL SETTINGS

Since DLG cannot be applied in VFL protocol, we test the performance of CAFE on various factors. We slice one image into 4 small pieces. Each worker holds one piece and the feature space dimension of each piece is $16 \times 16 \times 3$. The model is composed of 2 parts. The first part consists of 2 convolutional layers and 3 fully connected layers for each worker. The second part only consists of the softmax layer. In the training process, the pieces are sent into the first part respectively and turn to vectors as intermediate results. Local workers then exchange their intermediate results, concatenate them and put them into the second part. We set the batch size as 40 in VFL. Figure 4c and 4d show the CAFE loss curves and the PSNR curves on the three datasets in VFL cases. The data recovery is even better than the results in HFL. The PSNR values of CIFAR-10 and CIFAR-100 rise higher than 40. Same as the part in HFL, we put comparison with iDLG in Appendix C.

**Comparison with cosine similarity.** From Table 2b, we can conclude that the PSNR values still keep close to the ones by using CAFE. Scaling ambiguity in cosine similarity may also cause the drop in PSNR value. The performance gap between the loss of CAFE and the loss in Geiping et al. (2020) is much larger than the one in VFL, which indicates the utility of our auxiliary regularizers.

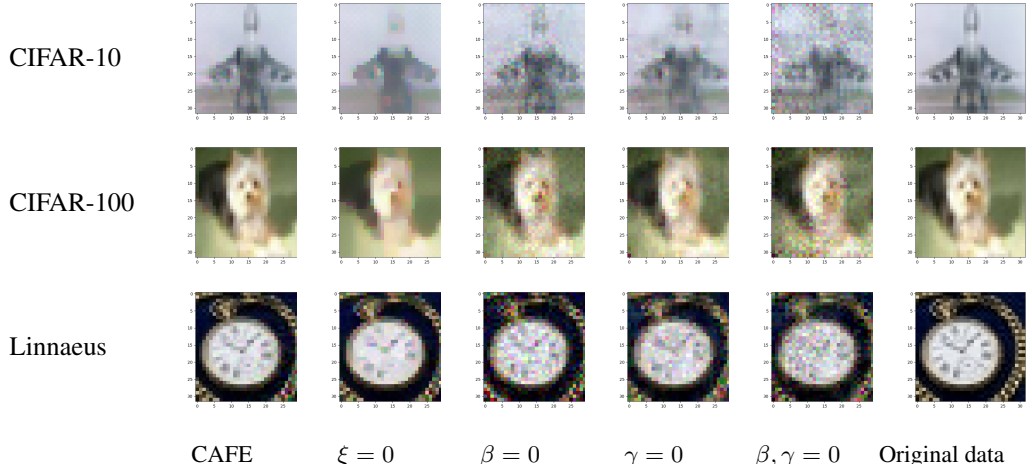

CIFAR-10

CIFAR-100

Linnaeus

CAFE     $\xi = 0$     $\beta = 0$     $\gamma = 0$     $\beta, \gamma = 0$     Original data

Figure 6: Effect of auxiliary regularizers

### 4.4 ATTACKING WHILE FL

Previous works have shown that DLG performs better on an untrained model than a trained one (Geiping et al., 2020). We also implement CAFE in the 'attacking while learning' mode, in which the FL process is ongoing. When the network is training, the selected batch data and the parameters of the model change every iteration, which may cause the attack loss to diverge. To address this issue, for each real data batch, we compute the real gradients and optimize the corresponding fake data $k$ times. We demonstrate on Linnaeus dataset, set $k = 10$ and stop CAFE after 1000 iterations (100 epochs). Figure 5 gives the curves of the training loss and the corresponding PSNR value. The PSNR value still can be raised to a relatively high value. It indicates that CAFE can be a practical data leakage attack in a dynamic training environment of FL.

### 4.5 ABLATION STUDY

We test CAFE under different batch size, batch ratio, and with (without) auxiliary regularizers.

**PSNR via Batch size.** Table 3 shows that the PSNR values still keep above 30 when the batch size increases with fixed number of workers and batch ratio. The result implies that the increasing batch size has little influence on data leakage performance of CAFE.

**PSNR via Batch ratio.** In HFL, 4 workers participate in the learning setting and we fix the amount of data held by each worker as 500. In the VFL case, we implement CAFE on a total of 800 images. In Table 4, we change the batch ratio from 0.1 to 0.01 while keeping the trained epochs as 800. For both settings, the data leakage performance keeps at the same level.

**Impact of auxiliary regularizers** Table 6 in Appendix D demonstrates the impact of auxiliary regularizers. From Figure 6, adjusting the threshold $\xi$ prevents images from being over blurred during the reconstruction process. TV norm can eliminate the noisy patterns on the recovered images and increase the PSNR. Images leaked without regularizing the Frobenius norm of the difference between the internal representations $\mathcal{Z}$ and $\hat{\mathcal{Z}}$ may lose some details and causes the drop of PSNR.

## 5 CONCLUSIONS

In this paper, we uncover the risk of *catastrophic data leakage in federated learning (CAFE)* through an algorithm that can perform large-batch data leakage with high data recovery quality. Extensive experimental results demonstrate that CAFE can recover large-scale private data from the shared aggregated gradients on both vertical and horizontal FL settings, overcoming the batch limitation problem in current data leakage attacks. Our advanced data leakage attack and its stealthy nature suggests practical data privacy concerns in FL and poses new challenges on future defenses.

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
