# OpenReview forum: "CAFE: Catastrophic Data Leakage in Federated Learning"
_ICLR.cc/2021/Conference — Reject_

### Official Review · AnonReviewer1 · 2020-10-26
**Simple, but unclear.**

**Rating:** 4
**Confidence:** 4

**Review:**

This work introduces CAFE, a novel training algorithm to leak training data in a federated learning setup. Extending from "deep leakage from gradient" fake images are optimised with respect to the difference observed from the client gradients (i.e. with the real images) and the one observed with the current version of the fake image. However, DLG does not work when the mini-batch size increases due to a messy gradient representation. In this work, the authors propose to keep track of the batch index. Indeed, it may happen that the server decides of the batch index corresponding to the training data that will be used by the client during the local training. Within such conditions, a malicious server can easily store fake images corresponding to specific indices and therefore optimise correctly each fake images w.r.t the corresponding real image.

It is clear from the obtained results that this method works, and that images are recovered. However, I am unsure about the relevance of the experimental protocol. 1. If the server does not ask for specific indices (and it is pretty common), the method is equivalent to DLG (i.e. does not work well) with large batches. 2. What if we don't have the gradients ? A common way of doing FL is to simply communicate the locally trained weights (with multiple local epochs). As specified in the introduction (point 3), the proposed method wouldn't work in this realistic scenario.

Then, I found Section 3.3 unclear is some aspects. Are the two proposed regularisation methods relying on the "real" image ? If so, isn't this a strong bias ? (we are not expected to have the input images). I suppose that this comes from the citation to the work of Geipinget al., 2020. However, these two paragraph should be re-written to clearly explain how we can extract the input vector and how it relates to eq. 8. "To promote the smoothness of the fake images, we assume the TV norm of the real images as a constant," -> We can't use the real image here, so it is not valid.

Pros:
+ In the given conditions, CAFE clearly outperforms the other approach to leak training data from gradients during FL.
+ Very simple attack to implement.

Cons:
- The conditions necessary to the success of the proposed methods seem to be quite strong and not really connected to a realistic FL framework.
- Small ideas can lead to drastic changes in the field, but the core idea of the paper is to solely store batch indices.

Remarks:
- "In this section, we provide necessary background of FL [in this section]."
- Figure 2 should be checked. "Aggreaged", "upload fake gradient" only once .
- What are t and b in Eq. 5.

---

> ### Author Response · Authors · 2020-11-21
> **We thank Reviewer 1 for detailed comments!**
>
> We thank the reviewer for the careful review and constructive feedback.
>
> •In VFL systems, since each local worker contains part and incomplete feature space of the data.  To successfully train the VFL model, it’s important to make sure the data features from each local worker are aligned according to the data indices.  As a result, the local workers must agree on the selected training data in each iteration,  which provides the server a chance to control the training data indices of each batch.
>
> •It’s a good point which we didn’t cover in the previous submission version. We proposed two methods to derive the multi-input vectors.  The first method is to use the sampling method and the second one is to build a linear equation system and solve the solutions which are the vectors we want.
>
> •We don’t use the real TV norm of the real data as the threshold ξ, we only estimate the average TV norm of the whole training dataset based on some other information as the threshold ξ.  Our simulation also indicates that an improper estimation may affect the performance of the algorithm.

---

### Official Review · AnonReviewer3 · 2020-10-28
**Review 3**

**Rating:** 4
**Confidence:** 3

**Review:**

The paper proposes an attack to extract information about training data from gradient updates sent as part of a federated learning setting.

The description of the attack setting and the attack algorithm is provided at a high level and detailed description is missing, making it hard to understand the novelty of the contribution. Experimental results do show that the attack is stronger than previous work. However, the overall presentation of the paper could be improved to be ready for a publication. Some suggestions listed below.

[Attack setting] It seems that the paper departs from some of the related work on information leakage by considering an attacker that can tamper with the federated learning process. Hence, the attacker is malicious and not benign. The authors should make this distinction clear if it is indeed the case. Commenting on why this malicious activity will not be noticed by the workers is important. For example, Figure 5 indeed shows that training accuracy is impacted by the attack.
How does this attack compare to the active attacker in the work by Melis et al?

The paper considers only 4 workers in most experiments. Federated learning usually has many more participants. Is this a problem? Do same workers need to be contacted at every iteration? Are there any assumptions on same data being used in each iteration.

[Attack algorithm]
The algorithm makes use of "data index alignment". Some guidance on what it means would facilitate the reading and understanding of the algorithm. Red part of Algorithm 1 should be expanded. How do these values get computed? Do they replace the blue parts or complement them?

[Presentation]
“As a motivating example, Figure 1 compares”: it would be best to motivate the algorithm key insight and not its improved performance over previous work that was already mentioned.
“However, as shown in Figure 2, a curious server can provide the same legitimate computation as a benign server while simultaneously perform data recovery in a stealthy manner. The server symmetrically generates fake” The figure does not describe how fake parameters are computed and it is not clear how pictorial representation shows this.

Minor details
Figure 1 captions: why 40 vs 10 x 4; “workers participating FL” -> workers participating in FL; please consider a better title for “4.4 ATTACKING WHILE FL”

---

> ### Author Response · Authors · 2020-11-21
> **We thank Reviewer 3 for detailed comments!**
>
> We thank all the reviewers for the positive comments and constructive feedback. We will correct typos, clarify some confusing points you mentioned, provide additional discussions on our assumptions. We summarize some common concerns below.
>
> •As we have mentioned in the paper, the server is a curious but honest attacker other than a malicious one which indicates that malicious attacks from the server will not occur during the data leakage process.  Various types of attacks are mentioned in Melis et al.  However, only some of which involving data leakage can be set as benchmarks and compared with CAFE. We will add a section discussing the attack comparing to some attacks in that paper.
>
> •It is common in HFL that a large number of agents join in the learning and many of them only participate in several rounds.  However, the more rounds the local agents participate in, the more likely the data will leak.  We will add another Table demonstrating the data leakage speed in CAFE comparing to DLG in VFL cases.  Table 1(a)and the newly added table indicate that the average required iterations on a certain image can be reduced to only 2-3 iterations. Moreover, a batch of data may contain both frequently trained data and less frequently trained data.  For the former ones, they may leak earlier.  We may regard these data as fully leaked data so that we just fix them and optimize the latter ones.
>
> •We will rearrange the figures and text description to make the presentation more clear and easy to understand.

---

### Official Review · AnonReviewer4 · 2020-10-28
**This paper studies the data leakage issue in federated learning, but lacks novelty in methodology and also details in some solutions.**

**Rating:** 3
**Confidence:** 5

**Review:**

This paper studies the data leakage issue in the federated learning. More precisely, when the servers have access to model parameters and gradients. It can recover the input data via gradient matching, and the authors claim that their method performs well even with large training batch sizes, e.g. over 40. Finally, the author also studies the possibility of attacking during learning, where they suggest that multiple updates of fake data helps. However, their contribution seems incremental, gradient matching is used in previous literature [zhu et al 2019], and their main modification is extra two regularization terms: total variation and internal representation regularization, and a data index alignment technique (whose exact meaning is unclear in the paper).

The following are some questions:

What does index alignment mean? Is that the server controls the indexes of samples chosen at each iteration? This seems to be very restrictive in practice, especially for horizontal federated learning.

Does the server have access to the aggregated grads from each worker separately or the workers aggregate all the gradients before sending them back to the server? The second scenario cab be achieved while secure aggregation technique.

In vertical federated learning, the gradients of part 1 of the network does not need to be exchanged with the server, as there is no average  operation needed, even the parameter itself does not need to be transferred to the server for the same reason, will your method work under this setting?

Some terms are not properly defined, such as normalized gradient descent, batch ratio, et. al.

Other questions:

What does the iterations represent in table 1a? Is that the number of iterations need to reach a 35 PSNR?

Using cosine dissimilarity decreases the PSNR, I assume this is because PSNR penalize the scale, is there noticeable degradation visually when using cosine dissimilarity?

In the attack during learning scenario, is there any intuition why optimizing fake data multiple times works better?

---

> ### Author Response · Authors · 2020-11-21
> **We thank Reviewer 4 for detailed comments!**
>
> We thank the reviewer for the constructive feedback.  Our response to your comments follow.
>
> •Since the server acts the role of an attacker, it has the access to the gradients before secure aggregation.  On the other hand, it will be an effective defense strategy if the server can only obtain the gradients after secure aggregation.
>
> •Although the gradients of some parts of the model don’t need to be uploaded and those parts of the model can be optimized locally.  However, the parameters of those parts of the model still need to be uploaded to the server.  Thus, the server can derive the gradients of those parts of the model by using the change of the parameter.  Even if the communication doesn’t need to occur in each iteration, the server can regard all the iterations between two communications as a big iteration in which the batch size is the sum of the batch size in every single iteration.  We will explain this point in the future version in detail.
>
> •The iterations represent in table 1a represents the average iterations for a single image to reach a 35 PSNR. We will add a similar table demonstrating the speed comparison between DLG and CAFE in VFL systems.
>
> •Once the PSNR value is above 30,  it will be hard for human eyes to tell the difference between the fake data and the real ones.   We can only measure the algorithm performance based on the PSNR value.
>
> •One method to help data leakage before the model is fully converged is to optimize the fake data multi times before the next batch of training data is chosen.  Since both the model parameters and the fake data are dynamic in the whole optimization process, it may be easier to help data leakage if we make the parameters relatively static compared to the fake data.

---

### Official Review · AnonReviewer2 · 2020-10-30
**Review for paper #995**

**Rating:** 4
**Confidence:** 2

**Review:**

The submission considers the problem of reconstructing private data from gradients in a Federated Learning system, which has been recently shown to a threat in distributed learning systems. Two types of federated learning systems are considered. Vertical federated learning (VFL) refers to the case where different agents hold different features of the same data points while  Horizontal federated learning (HFL) refers to the case where different agents how all the features of different subsets of the data.

Previous attacks solve an optimization problem that aims to infer the data by minimizing the mismatch between real gradients and fake gradients. This method suffers difficulty when the number of samples in one round is large. The paper proposes CAFE, which takes advantage of the fact that in vertical federated learning (VFL) systems, the server can identify the indices of the samples that are selected in each round. This extra information help reduce unwanted solutions in the optimization problem and help improve the reconstruction performance. The authors conduct experiments to show that the proposed algorithm outperforms previous works.

The paper shows that data leakage from gradients is a potential threat in VFL systems even when the batch size is large. However, I have the following concerns about the paper.

1. The paper claims that the attack also works for the HFL setting. However, this is no well justified for the following reasons:
(1) The assumption that the server knows the indices of the samples that are selected in each round is not valid in general for the HFL setting since each agent can sample a batch locally.
(2) In HFL settings, it is generally assumed that the number of agents is large and each agent only participates in a few rounds, which is not considered in the experiments in the submission.
2. In the experiments, it is shown that as the number of training epochs grows, better inference on the private data can be made. It would be better if the authors can also include the training error on each epoch in the same plot. It is believable that if the training goes on forever,  enough information can be inferred about the training samples. However, it might be good to see whether the server can infer the train samples before the model has already converged.

I hope the authors can address my above concerns in the response.

---

> ### Author Response · Authors · 2020-11-21
> **We thank Reviewer 2 for detailed comments!**
>
> We thank the reviewer for the constructive feedback.  We will correct typos,  clarify some confusing points you mentioned, provide additional discussions on our assumptions. We address some comments below:
>
> •The assumption that the server knows the sample indices may be strong in HFL set-tings.   However,  we have already demonstrated that  CAFE  is an effective attacking method in VFL settings.  To make the simulation results more convincing, we will add more simulations based on VFL systems.
>
> •It is common in HFL that many of them only participate in several rounds.  However, the more rounds the local agents participate in, the more likely the data will leak.  We will add another Table demonstrating the data leakage speed in CAFE comparing to DLG in VFL cases.  Table 1(a) and the newly added table indicate that the average required iterations on a certain image can be reduced to only 2-3 iterations.
>
> •Firstly,  we will add another figure to demonstrate the simulation of attacking while federated learning in VFL which will make the simulation more convincing.  A successful CAFE attacking will reconstruct the data before the model has converged, which is similar to the result we shown in Figure 5.

---

### Author Response · Authors · 2020-11-21
**General response**

We thank all the reviewers for the positive comments and constructive feedback.  We will correct typos, clarify some confusing points you mentioned, provide additional discussions on our assumptions.  We summarize some common concerns below.

•The assumption that the server knows the sample indices may be strong in HFL set-tings.   However,  we have already demonstrated that  CAFE  is an effective attacking method in VFL settings.  To make the simulation results more convincing, we will add more simulations based on VFL systems.•The term index alignment means that in each iteration, we select the fake data whose data indices are the same as the ones of the selected training data.  We will make it clear in the future submission.

•It is indeed common in  FL  that local workers communicate locally trained weights instead of gradients.  Thus,  the server can derive the gradients of those parts of the model by using the change of the parameter.  Even if the communication doesn’t need to occur in each iteration, the server can regard all the updates between two consecutive communications as a big iteration in which the batch size is the sum of the batch size in every single iteration.  We will explain this point in the future submission in detail.

•Our idea is simple and small.  However, it may cause catastrophic data leakage which may lead to grave consequences.   That’s the reason why we think the algorithm is worthy to be proposed and we want to remind people of some unsafe and neglected trivialities in federated learning such as batch indices.

---

### Decision · Program_Chairs · 2021-01-07
**Final Decision**

**Decision:**

Reject

**Comment:**

This paper focuses attacks on federated learning. The reviewers had the following concerns:
- The assumption of knowledge of batch indices is unrealistic in an HFL setting
- The setup only works when doing a single epoch (I believe the authors claim that it is applicable in more general settings, but evidence to that effect has not been provided)
- The novelty of the approach is somewhat limited.
- The description of the algorithm and comparison to prior work could be clearer.

I raised the question of whether the reviewers would be more positive if there were no claimed results on HFL. They still did not seem positive enough to justify acceptance (due to the other reasons mentioned above).